# The Role of Mitochondria in the Mechanisms of Cardiac Ischemia-Reperfusion Injury

**DOI:** 10.3390/antiox8100454

**Published:** 2019-10-06

**Authors:** Andrey V. Kuznetsov, Sabzali Javadov, Raimund Margreiter, Michael Grimm, Judith Hagenbuchner, Michael J. Ausserlechner

**Affiliations:** 1Cardiac Surgery Research Laboratory, Department of Cardiac Surgery, Medical University of Innsbruck, Innsbruck A-6020, Austria; michael.grimm@tirol-kliniken.at; 2Department of Pediatrics I, Medical University of Innsbruck, Innsbruck A-6020, Austria; 3Department of Physiology, School of Medicine, University of Puerto Rico, San Juan, PR 00936-5067, USA; sabzali.javadov@upr.edu; 4Department of Visceral, Transplant and Thoracic Surgery, Medical University of Innsbruck, Innsbruck A-6020, Austria; raimund.margreiter@tirol-kliniken.at; 5Department of Pediatrics II, Medical University of Innsbruck, Innsbruck A-6020, Austria; judith.hagenbuchner@i-med.ac.at

**Keywords:** heart, ischemia-reperfusion, cytoskeleton, energy metabolism, mitochondria, mitochondrial heterogeneity, preconditioning, reactive oxygen species, signaling

## Abstract

Mitochondria play a critical role in maintaining cellular function by ATP production. They are also a source of reactive oxygen species (ROS) and proapoptotic factors. The role of mitochondria has been established in many aspects of cell physiology/pathophysiology, including cell signaling. Mitochondria may deteriorate under various pathological conditions, including ischemia-reperfusion (IR) injury. Mitochondrial injury can be one of the main causes for cardiac and other tissue injuries by energy stress and overproduction of toxic reactive oxygen species, leading to oxidative stress, elevated calcium and apoptotic and necrotic cell death. However, the interplay among these processes in normal and pathological conditions is still poorly understood. Mitochondria play a critical role in cardiac IR injury, where they are directly involved in several pathophysiological mechanisms. We also discuss the role of mitochondria in the context of mitochondrial dynamics, specializations and heterogeneity. Also, we wanted to stress the existence of morphologically and functionally different mitochondrial subpopulations in the heart that may have different sensitivities to diseases and IR injury. Therefore, various cardioprotective interventions that modulate mitochondrial stability, dynamics and turnover, including various pharmacologic agents, specific mitochondrial antioxidants and uncouplers, and ischemic preconditioning can be considered as the main strategies to protect mitochondrial and cardiovascular function and thus enhance longevity.

## 1. Introduction

Mitochondria are the main source of ATP production under the aerobic conditions necessary for normal cell function and viability. In addition to ATP synthesis, mitochondria regulate a wide range of metabolic processes and signaling pathways in the cell (Figure 1A). They synthesize different metabolites, regulate cellular redox potential and play an important role in ion regulation, in particular, in Ca^2+^ homeostasis, thermogenesis, and programmed cell death (apoptosis) [1,2,3,4,5]. Mitochondria actively participate in cellular Ca^2+^ signaling [6,7,8,9,10]. A crucial role of mitochondria has been established in many aspects of entire cell physiology and pathophysiology in a broad spectrum of diseases including heart and brain ischemia-reperfusion (IR) injury, heart failure (HF), inherited diseases, diabetes, obesity, toxicology, side effects of pharmacological treatments and other pathological conditions as well as in aging [11,12,13,14,15,16,17,18,19,20]. For example, significant impairment in energy metabolism and mitochondrial function has been demonstrated in the skeletal muscle during diabetes and obesity [21,22].

Mitochondrial dysfunction causes cell/organ injury through several mechanisms (Figure 1B), including diminished cellular energy status (low cellular ATP level, energy stress), enhanced production of reactive oxygen species (ROS) including superoxide anions, hydrogen peroxide (H_2_O_2_), hydroxyl radicals (OH^.^) and peroxynitrite [16] with the subsequent development of oxidative stress. Furthermore, mitochondrial damage is associated with the release of several apoptosis activated factors, leading to programmed cell death [4,23,24,25]. Disturbances in ionic balance, particularly an increase in mitochondrial and cytoplasmic Ca^2+^, stimulates mitochondrial permeability transition (PT) accompanied by the opening of non-selective channels known as the PT pores (PTP) that allow free movement of ions and other solutes with a molecular mass <1.5 kDa across the inner mitochondria membrane (IMM). As a result, PTP opening enhances colloid-osmotic pressure in the matrix, leading to mitochondrial swelling associated with the activation of proteases and lipases that eventually lead to cell death and permanent loss of cardiomyocytes in the heart [26,27,28]. In addition, decreased mitochondrial function leads to a low level of cellular ATP, together with elevated Ca^2+^, resulting in cardiomyocyte super-contracture, disruption of plasmalemma and therefore necrotic cell death [28]. However, due to the complex relationship between decreased cellular ATP level and increased ROS and Ca^2+^, precise molecular mechanisms and consequences of these events are not completely understood. So, the relationship between organ dysfunction and mitochondrial impairment is not simple and certainly is not limited to the failure in ATP production. Rather, mitochondrial damage can affect cell viability in several ways, including different signaling mechanisms that can co-exist simultaneously in the same cell and communicate with each other in response to specific stimuli.

Mitochondria may be separate or found in a network, where permanent dynamic fission and fusion can occur. Moreover, specific mitochondrial quality control may use the interplay between fusion and fission, removing damaged or incorrect organelles. This selective autophagy of damaged or defective mitochondria (mitophagy) can be dependent on their low mitochondrial membrane potential (ΔΨ_m_) [29,30,31,32,33]. Mitochondria play a key role in the pathogenesis of cardiovascular diseases such as IR injury, loss of cardiomyocytes, HF and various cardiomyopathies [12,13,14,15,16]. They are central in the induction of apoptotic and necrotic cell death associated with the accumulation of ROS which causes oxidative stress and cell injury due to protein, lipid and DNA oxidation, although at low concentrations (under physiological conditions), ROS participate in cellular signaling [34,35,36]. Thus, new pharmacological agents and conditional strategies (e.g., ischemic preconditioning and postconditioning) designed to modulate/stabilize mitochondria can provide effective therapeutic approaches to prevent cell/organ dysfunction in response to pathological stimuli. This is especially important for organs with high energy demands such as the heart, where mitochondria occupy about 35% of the volume of adult cardiomyocytes and provide about 90% of ATP through oxidative phosphorylation (OXPHOS). 

Modern scientific technologies in cellular and mitochondrial research remarkably improve our ability to elucidate molecular mechanisms of cardiac function and cellular longevity as well as the role of mitochondria in cell dysfunction. As mentioned above, mitochondria are directly involved in the pathophysiological mechanisms of IR injury. Although restoring blood flow and tissue reoxygenation after myocardial ischemia can partially recover cardiac function, it also induces additional (up to 50%) damage known as “reperfusion injury” due to excessive ROS production and Ca^2+^ overload [37]. Therefore, it is critically important to improve the recovery of organ function and reduce injury at reperfusion. Both the ischemic and reperfusion phases are associated with the obliteration of the cellular/mitochondrial energy production necessary for cardiac contractile function. Mitochondria interact with the environment and hence their function can be modulated depending on the concentration of growth factors, oxygen, ATP, ROS and Ca^2+^ in the cytoplasm and matrix [35]. Mitochondria play a key role in maintaining Ca^2+^ homeostasis through spatial and functional interaction with both cytoplasm and sarcoplasmic reticulum (SR) Ca^2+^ [6,7,8,9,10]. Disturbances in Ca^2+^ homeostasis are known to play a central role in the pathogenesis of cardiac dysfunction. However, the mechanisms regulating mitochondrial ROS (mitoROS) production and Ca^2+^ homeostasis as well as the crosstalk between these two processes remain unknown. Future studies aim to develop new pharmacological substances that can selectively target mitochondrial damage and restore the functional capacity of the heart through the improvement of mitochondrial metabolism and function. Here, we summarize and discuss the main regulatory aspects of mitochondrial physiology: function, intracellular organization, dynamics, and the role of mitochondrial interactions with other cellular systems such as energy transfer systems, the cytoskeleton and the SR. Also, we discuss the role of mitochondria in cardiac dysfunction during coronary heart diseases, particularly focusing on cardiac IR injury.

## 2. An Overview of the Techniques Used for the Analysis of Mitochondrial Function, Dynamics and Intracellular Organization

The analysis of mitochondrial function/dysfunction is important in the study of the mechanisms of mitochondria-mediated IR injury, as well in the diagnosis and therapy.

Regulation of mitochondrial function is one of the fundamental problems in understanding cell/organ energy metabolism and cellular bioenergetics. Mitochondrial respiratory function reflects the capacity for aerobic energy production, which is more indicative of organ function/viability and injury than a simple assessment of the cellular ATP levels [18,19]. Therefore, the analysis of mitochondrial function is extremely important in basic research of mitochondrial physiology and clinically oriented studies, as well as in the diagnosis of various metabolic diseases, including cardiac IR injury. Usually, mitochondrial respiratory function is analyzed by routine oxygraphy by measuring the rate of oxygen consumption in isolated mitochondria in vitro, permeabilized muscle fibers or cells in situ or in intact cells in vivo [38,39,40,41]. In addition, organ heat release can be measured as an indicator of myocardial energy metabolism activity and its changes after IR [42]. Analysis of mitochondrial respiratory function in situ in permeabilized preparations of cells or muscle fibers has a number of serious advantages as various artifacts of mitochondrial isolation can be avoided and important contacts with other cellular systems (cytoskeleton, SR) can be preserved [38,39,40,41]. Importantly, the technique requires a small quantity (15–20 mg) of biological material and it can be successfully used when samples with limited size are available, for example in human biopsies or transgenic animals.

The combination of high-resolution respirometry and fluorescent confocal imaging of mitochondria can be optimal for detailed studies of mitochondrial function, dynamics, and regulatory pathways, as well as metabolic and functional changes during cell/organ dysfunction. Further development of such complex analysis represents a current challenge for biomedical research. Confocal imaging studies of changes in mitochondrial function, arrangement, morphology, ROS/Ca^2+^ and ΔΨ_m_ using specific fluorescent probes, significantly help to analyze the time course of ROS and mitochondrial Ca^2+^ changes and the consequences of events during reversible or irreversible IR injury [43,44]. A deeper insight into the sequence of pathological episodes which lead to organ damage under IR is important to define sites and approaches for therapeutic intervention and protection.

The various imaging approaches, in contrast to in vitro analyses of isolated mitochondria, are well suited to study mitochondria in different subcellular compartments as well as complex mitochondrial dynamics and organization, which are critical for understanding the role of various mitochondrial properties, functions and networks in the cell [43,44,45]. Confocal microscopy, using various mitochondria-specific fluorescent probes, together with the autofluorescence of mitochondrial flavoproteins and NADH, was widely used for the imaging and characterization of mitochondria in situ, in permeabilized muscle fibers/cells, or in vivo in intact cells [44,45,46,47,48,49,50]. Importantly, mitochondrial confocal imaging can not only visualize mitochondrial intracellular arrangement, morphology, dynamics, networks, and heterogeneity but also quantitatively analyze mitochondrial redox state, ΔΨ_m_, ROS, and Ca^2+^. This provides a suitable way to detect and study many structural and functional changes, damage or defects in mitochondrial and metabolic pathologies, including IR injury. Also, mitochondrial green fluorescent proteins specifically targeted to mitochondria, fluorescence resonance energy transfer (FRET) and beam-scanning multifocal multiphoton (4Pi)-confocal and stimulated emission depletion (STED) microscopy [51] can be helpful in the investigation of specific protein–protein interactions, conformational changes and mitochondrial interactions with other cellular systems (i.e., cytoskeleton and ER). 

## 3. Mitochondrial ROS (mitoROS)


*MitoROS are Linked to the Consequences and Physiological Effects of IR Injury*


One of the main cellular ROS producers are mitochondrial respiratory chain complexes, but, at the same time, mitochondria and mitochondrial membranes can be significantly damaged by ROS over-production generated either by mitochondria themselves or by other cellular sources of ROS [52,53,54]. Therefore, pharmacological protection against cellular injury requires targeting of mitochondria. MitoROS generation is tightly linked to the cellular redox state, and ROS are central to the general cellular metabolism, also playing an important role in cellular signaling [36,55]. Also, increased ROS production can be directly linked to increased Ca^2+^ and the induction of apoptosis. The mitochondrial respiratory chain complexes I–IV transfer electrons to oxygen, producing superoxide radicals as a byproduct of this process due to the incomplete reduction of oxygen. Complexes I and III are considered as the main producers of mitochondrial superoxide radicals [17,56,57,58,59,60,61,62], which then can be converted to hydrogen peroxide by mitochondrial superoxide dismutase (SOD), which in turn can be scavenged in catalase reactions [17]. At low concentrations, ROS can mediate the physiological effects; however, the overproduction of ROS is involved in the pathogenesis of heart coronary diseases including IR injury.

Thus, the deterioration of both mitochondrial Ca^2+^ and mitoROS are responsible for triggering cell death/loss during cardiac IR. It has been found that in cardiac and other cells types, biphasic mitoROS dynamics may occur, which include gradual mitoROS increase followed by mitoROS flash [63,64,65]. Also, such a flash can be initiated by an external ROS, which is a well-known and important phenomenon of ROS-induced ROS-release, first described by Zorov et al. in cardiomyocytes [63]. In our very recent study, we have also demonstrated similar mitochondrial ROS-ROS communications/interactions and ROS flashes (self-amplifying ROS bursts) in several cancer cell lines, where these effects were very heterogeneous but always in parallel with mitochondrial Ca^2+^ sparks and severe depolarization (dramatic drop in ΔΨ_m_) of these organelles [47]. However, the exact interplay between different ROS, as well as the effects of ROS produced by different cellular sources (including mitochondrial respiratory chain complexes), is still not clear.

## 4. Mitochondrial Dynamics: Fission/Fusion and Motility


*Mitochondria are Dynamic and Well Organized Organelles in the Cell*


Importantly, the intracellular mitochondrial position/arrangement, morphology, heterogeneity and dynamics (fission/fusion, motility) are tightly regulated by several important proteins [66,67,68,69,70,71,72,73,74]. All these processes present an important part of general mitochondrial physiology and may significantly change under various pathologies and diseases [68,69,70], including IR, various effects of ROS, and during programmed cell death (in particular, mitochondrial fragmentation can be the first sign of apoptosis induction) [75,76]. In many cases, mitochondrial motilities can be very important for energy production, as well as Ca^2+^ regulation in specific cell regions, and this process can be injured in certain diseases. Importantly, pathology-associated changes in mitochondrial morphology/dynamics can be strongly linked to programmed cell death [69,75,76,77]. Also, myocardial infarction was found to be associated with changes in the balance between actions of the fission and fusion proteins in rats [70,78,79].

Mitochondrial fusion and fission, together with mitophagy (removal of defected mitochondria) and mitochondrial biogenesis/turnover, are important components of mitochondrial quality control [29,30,31,32,33]. The mechanism of removal of defective mitochondria is mostly based on their low ΔΨ_m_. While fusion allows for the exchanging of mitochondrial matrix content between normal and defective mitochondria (e.g., redistribution of mtDNA), further fission will produce a normal mitochondrial population again, therefore supporting a repair mechanism. This may also play a role in the protection against IR injury [29,80]. It has been suggested that autophagy can be activated in prolonged ischemia and reperfusion [80,81] or during hypoxia–reoxygenation injury [82]. The lack of proper quality control of mitochondria and the enlargement of mitochondrial defects can be part of the mechanism of organ injuries.

## 5. The Role of Cytoskeleton Proteins in the Regulation of Mitochondrial Function


*Cytoskeletal Elements are Involved in the Control of Mitochondrial Respiratory Function*


Notably, cellular function, intracellular arrangement/organization and the morphology of mitochondria are frequently defined by the specific internal cell structure [83] and mitochondrial communication with the sarcoplasmic reticulum and certain cytoskeletal elements [84,85,86], such as tubulin beta (Figure 2, [86]) and specific isoforms of the cytolinker protein, plectin [87,88]. For example, in the skeletal muscle of conditional plectin knockout mice (MCK-Cre/cKO), mitochondrial content was reduced; mitochondria were aggregated in the sarcoplasmic and subsarcolemmal regions, and were no longer associated with Z-disks [88]. The communication of the mitochondria–cytoskeleton elements (tubulin and plectin) is suggested to include their tight structural connections with Mitochondrial Voltage-Dependent Anion Channel (VDAC) [84,85,86,87,88,89,90,91] and, therefore, they can actively manipulate the outer mitochondrial membrane permeability to ADP and other important metabolites [91]. Possibly, some other cytoskeletal proteins such as desmin [84] and vimentin [90] can also be involved here.

In summary, recent data shows that the tubulin beta II isoform and the plectin 1b isoform can be involved in the control of fluxes via the energy transferring super-complex—VDAC, mitochondrial creatine kinase (mitCK) and ATP-ADP translocase (ANT), regulating mitochondrial respiratory function, cell bioenergetics and therefore entire cellular physiology.

## 6. Mitochondrial Heterogeneity and Subpopulations: Possible Physiological and Pathophysiological Roles

 *Mitochondria are, in Many Aspects, Heterogeneous in the Cell*

Confocal fluorescent imaging of mitochondria in cardiac tissue or in isolated cardiomyocytes has revealed that mitochondria are either clustered or arranged in a highly organized manner. Some specific features of mitochondrial function and, in particular, their communication with other structures can be considered as an important component in the mechanism of apoptosis transmission [92]. Importantly, mitochondria localized in different compartments of the cell can have different morphologies and biochemical properties, so they can be rather heterogeneous in various structural and functional aspects under normal [45,93,94,95,96,97,98,99,100,101] and pathological [18,43,46,97,102] conditions. 

The main mitochondrial function (ATP production), their shape/morphology, redox state, also mitochondrial ROS and Ca^2+^, as well as other important properties of mitochondria can be dependent on the numerous communications of these organelles with the rest of cell, including mitochondrial “signaling in” and “signaling out” phenomena (Figure 3).

Intracellular ADP and oxygen gradients, as well as a local and region-specific increase in cellular Ca^2+^, may also contribute to the control of mitochondrial function, dynamics, and morphology, thus playing an important role in the formation of mitochondrial heterogeneity. Moreover, the specific role of the p66Shc protein in the cellular mechanisms leading to ROS increase, oxidative stress and activation of programmed cell death by apoptosis [103] during IR and other pathologies has been demonstrated [104,105,106]. The involvement of various protein kinase C (PKC) isoforms during redox stress (differing in their biochemical properties and sensitivities) produces a complex pattern of PKC signaling (Figure 3, [104]) potentially also contributing to mitochondrial heterogeneity.

All these processes should be taken into account considering the general basis for the well-known phenomenon of mitochondrial heterogeneity found in many cell types such as cardiomyocytes, hepatocytes, human umbilical vein endothelial cells (HUVEC), astrocytes and various human carcinoma cells, which can be associated with (or served for) mitochondrial region-specific functions [93,94,95,100] (Figure 4A). 

Cardiac cells contain discrete pools of mitochondria known as perinuclear (PNM), intermyofibrillar (IFM) and subsarcolemmal (SSM) mitochondria (Figure 4B) with different functions, shape, absolute size and internal cristae arrangement [99,107]. Notably, these mitochondrial subpopulations may not only differ by morphology and biochemical properties, but they may also have different region-specific specializations depending on their intracellular localization/environment and particular cellular demands. Most importantly, mitochondrial subpopulations may be differently involved in pathological processes like IR injury [108] and various cardiomyopathies [109], showing their different sensitivity to injury. It can be suggested that distinct mitochondrial subsets, clusters, or even single mitochondrion may perform diverse tasks for specific cellular requirements [93,94,100,110]. By monitoring (using fluorescent imaging) flavoprotein autofluorescence (fluorescent only in the oxidized state), a higher oxidation of SSM was shown [45,46,101]. Similar phenomena have been demonstrated for rat soleus and gastrocnemius muscles, where a higher oxidative state correlated with elevated mitochondrial Ca^2+^ (monitored by Rhod-2) [45]. The heterogeneity of mitochondrial Ca^2+^-induced PTP induction has also been studied in brain mitochondria [102]. At the same time, PNM subsets (Figure 4B) may generate ATP close to the nucleus for nuclear import [110,111] and for a variety of other nuclear functions. 

Thus, various mitochondrial subpopulations are present in the cell that may be differently involved in physiological and pathological processes, clearly demonstrating mitochondrial heterogeneity.

Exposure of cells loaded with the ΔΨ_m_-specific probe tetramethylrhodamine methyl ester (TMRM, red) and 2,7-dihydrodichlorofluorescein diacetate (DCF-DA, green) to laser irradiation activates extensive mitoROS production, detected as a strong increase in DCF fluorescence, together with a collapse of mitochondrial ΔΨ_m_, visible as a strong decrease in TMRM fluorescence (the appearance of green mitochondria). This effect can be used as a convenient tool for mitoROS generation and the induction of photo-oxidative stress and Ca^2+^ transients. Heterogeneity of mitoROS and ΔΨ_m_ has been demonstrated in various cells during IR, oxidative stress and photo-oxidative stress [112] (see Figure 5).

Taken together, the study of the mitochondrial heterogeneity may thus represent a new challenging area in mitochondrial and cellular physiology.

### 6.1. Heterogeneity of Mitochondria in Pathology

Mitochondrial defects can also be heterogeneously distributed due to the phenomenon of their mosaic expression and existence of metabolic gradients and micro-compartmentation. The heterogeneity of the mitochondrial redox state, ΔΨ_m_, and Ca^2+^ have been studied in cardiac cells under pathological conditions. It has been shown that mitochondrial defects can be heterogeneously distributed and may have a different degree of damage in distinct mitochondrial subpopulations [18,108,109,113,114]. Also, morphological alterations of mitochondria and myofibrils are not uniformly distributed in the ischemic zone, showing a striking heterogeneity in the extent of IR damage. This is in accordance with the fact that, in the heart, the ischemic injury does not evolve in a uniform manner and regional differences in metabolism and energy requirements may exist in the myocardium [113] where large metabolic perturbations are expected. It is known that myocardium injury and tissue necrosis usually originate in the endocardium and, with time, may migrate as a “wave front of cell death” towards the epicardial surface.

The intrinsic heterogeneity of mitochondria includes the existence of subpopulations with different biochemical and morphological properties, possibly due to differences in metabolism and energy requirements in various cell regions. SSM and IFM populations have been obtained in skeletal and cardiac muscles by selective isolation procedures [99]. Moreover, different functional (redox state) behavior of these mitochondrial subpopulations was observed in *in situ* mitochondria [101]. Mitochondrial subpopulations may be differently involved in physiological and pathological processes including cardiomyopathy, apoptosis and normothermic IR injury [108,109,114]. Also, it has been shown that substrate (i.e., glucose, serum, growth factors) deprivation may increase the subcellular heterogeneity of mitochondrial energization in intact cells [35,44]. Heterogeneous damage of mitochondria may be a result of heterogeneous oxygen, Ca^2+^, or ROS distribution in the ischemic cell, or it can be secondary to heterogeneous mitochondrial functioning, due to heterogeneity in redox state, Ca^2+^ and ΔΨ_m_ (see Figure 3). Analysis of the functional/structural diversities of mitochondria may therefore be important in the study of the mechanisms of cardiac IR injury.

### 6.2. Mitochondrial Heterogeneity and Apoptosis

It is well known that a component of the mitochondrial respiratory chain, cytochrome *c*, together with other pro-apoptotic factors, participate in the mechanism of apoptosis for the formation of the apoptosome. The release of cytochrome *c* from mitochondria decreases mitochondrial respiration and thus ATP production. However, ATP is needed for apoptosis at several sites. Thus, it can be suggested that the cytochrome *c* derived from one mitochondrion will support apoptosis, while cytochrome *c* not released will further support oxidative phosphorylation (and ATP), demonstrating its possible heterogeneity. This phenomenon has been suggested and obliquely shown in heart preservation, transplantation and reperfusion, and in cardiac cold ischemia-reperfusion injury (CIR) [18]. Heterogeneous mitochondrial damage has also been shown more directly by fluorescent confocal microscopy [43,45,99].

Direct imaging of the mitochondrial functional state in permeabilized myocardial fibers from rat hearts is able to demonstrate flavoprotein autofluorescence as an indicator of mitochondrial redox state, mitochondrial Ca^2+^ from the fluorescence of Rhod-2 and ΔΨ_m_ from TMRE fluorescence. This imaging was compared between control fibers and after cold ischemia (organ preservation), transplantation and reperfusion, the conditions that produce a complex pattern of multiple damages. In controls, the regular mitochondrial arrangement typical of cardiomyocytes was clearly seen, and relatively homogeneous fluorescence of mitochondrial flavoproteins and the specific mitochondrial Ca^2+^ indicator Rhod-2 showed homogeneity of mitochondrial redox state and Ca^2+^ content. Similarly, imaging of TMRE fluorescence demonstrated a homogeneous pattern of ΔΨ_m_. After CIR, myocardial fibers showed heterogeneity of redox states of mitochondria and numerous “black holes” in Rhod-2 fluorescence, indicating mitochondria that lost Ca^2+^ (more clearly visible as green spots in the merge image). Moreover, “black holes” in TMRE fluorescence and spots with only green flavoprotein fluorescence in merge images show depolarized mitochondria (collapse of ΔΨ_m_) and localized PTP opening after CIR [43].

All these effects may be associated with heterogeneous cytochrome *c* release, leading to heterogeneous mitoROS generation and mitochondrial permeability transitions [18,43]. However, the development and role of apoptosis in CIR (organ preservation for transplantation) of the myocardium is still unclear. Confocal imaging of mitochondria allows for the topological assessment of mitochondrial defects, providing new insights into the mechanisms of cardiac IR injury, demonstrating spatial and temporal heterogeneity in mitochondrial redox potential and ΔΨ_m_ including local transients and propagated metabolic waves. Imaging of mitochondria allows topological assessment of mitochondrial defects, therefore providing new insights into the mechanisms of the cardiac IR injury.

## 7. The Role of Mitochondria in Cellular Signaling and The Role of Kinase Signaling Pathway


*Mitochondria Actively Participate in Cellular Signaling*


Mitochondria communicate with the rest of the cell using numerous pathways and second messengers. These organelles came to be considered an integral part of multiple cellular signaling cascades (see Figure 3 and [3,34,35,36,63,92,115,116,117,118]). It has been shown that C-Raf kinase can form a complex with mitochondrial VDAC in vivo, blocking in vitro reconstitution of VDAC channels in bilayer membranes. It was suggested that the C-Raf (Figure 3) interaction with VDAC may play a role in the Raf-induced inhibition of cytochrome *c* release from mitochondria, as well as in regulating mitochondrial function [116,117]. More recent results have demonstrated that some ligands to VDAC, e.g. erastin, which binds to VDAC2, alters the permeability of the outer mitochondrial membrane (OMM) and may induce non-apoptotic cell death selectively in tumor cells harboring activating mutations in the RAS–RAF–MEK pathway (RAS is a product of the KRAS2 gene). However, whether this can also be associated with changes in the permeability of VDAC for ADP (and sensitivity of mitochondria to ADP in situ) is not known.

A direct link between the expression of oncogenic RAF and alterations in mitochondrial matrix Ca^2+^ and ROS levels has been demonstrated [35]. The studies demonstrated that the RAS–RAF–MEK–extracellular signal-regulated kinase (ERK) signaling pathway, protein kinase B (Akt), and Bcl-2 family proteins (Figure 3) actively participate in regulating mitochondrial Ca^2+^ and ROS [35]. Mitogen-activated protein kinases (MAPKs) including ethanolamine kinase (ETK1/2), p/38, and c-Jun N-terminal kinase (JNK) are thought to exist downstream of the Src–PKC signaling module, although the role of MAPK remains undetermined. This mechanism involves the redox-sensitive activation of transcription factors through PKC and tyrosine kinase signal transduction pathways.

## 8. Mitochondrial Energy Metabolism in Cardiac IR Injury


*Mitochondria are Vitally Involved in the Molecular Mechanisms of Cardiac IR Injury*


Mitochondria can be significantly damaged during both prolonged normothermic or cold (during organ preservation) IR injury [18,19,26,118,119,120,121,122,123]. Mitochondrial dysfunction plays a key role in the pathogenesis of this injury and various other heart pathologies [12,19,124]. Both mitochondria and the energy transfer networks may deteriorate under pathological conditions, leading to severe organ injury. In IR injury, the lack of oxygen and respiratory substrates stops OXPHOS, leading to collapses of ΔΨ_m_, swelling of mitochondria, Ca^2+^ overload, cytochrome *c* release, disruption of cellular membranes and finally cell necrosis [28]. Thus, mitochondria play central roles in both types of cell death: necrosis and apoptosis. 

Reestablishing blood flow and reoxygenation of the tissue can restore organ function, but leads to cardiac tissue/organ damage due to ROS production, oxidative stress and reperfusion injury. Increased production or insufficient elimination of toxic ROS by mitochondria upon reperfusion leads, in turn, to subsequent peroxidation of proteins and mitochondrial respiratory complexes and phospholipids; for example peroxidation of cardiolipin required for complex III and complex IV activities [125,126,127]. In particular, respiratory complex I can be damaged in IR due to oxidation of SH groups. At the same time, complex I can intensify its ROS production during IR injury [123]. Furthermore, impairment of intracellular Ca^2+^ homeostasis, PTP opening, ΔΨ_m_ loss, cytochrome *c* release, apoptosis and modification of DNA are associated with, and/or are the consequences of IR injury [120,123,128,129,130,131].

Among the main mechanisms that underlie mitochondrial dysfunction in IR injury are cardiomyocyte death/loss, cellular Ca^2+^ dysregulation and ATP depletion, the release of proapoptotic proteins, and induction of oxidative stress by the mitochondrial transition to ROS generation immediately after reperfusion/reoxygenation [129]. However, the complex interrelationships between mitoROS, ΔΨ_m_ and Ca^2+^ are not completely understood. On the other hand, ROS contribution to organ injury may be remarkably dependent on the capacity of cellular antioxidant systems shown to be reduced in pathologies.

Several factors were suggested to contribute to mitochondrial injuries, such as altered phospholipid composition (especially of the IMM) and an excess of the long-chain fatty acid, CoA. Mitochondrial DNA has an increased mutation level; it has a lower repair ability, less protection against oxidative stress and an increased level of ROS, and therefore may have more ROS damage. However, some ROS formation in low (sub-lethal) concentrations is critical for cellular signaling [35,132]. Damaged complexes I and III are thought to be the major sources for mitochondrial free radical production [56,57,58,59,60]. In addition, it has been proposed that ROS generation after IR injury can also stimulate an inflammatory response [133].

The most energy-consuming organ, the heart, containing the biggest mitochondrial content (30% of cell volume), is also most sensitive to IR injury compared with other, less energy-dependent organs [134]. Damage to mitochondrial respiratory chain complexes after ischemia (including also cold ischemia and organ preservation) alone is also significantly different and produces more pronounced injury after reperfusion. In a rat heart model of cold ischemia, heart transplantation and 24 h of reperfusion (CIR) and organ preservation, post-ischemic reperfusion resulted in a dramatic decline in NADH-linked ADP stimulated respiration due to specific damage to respiratory complex I. Similar correlations were found for succinate (complex II) and tetramethyl-1,4-phenylendiamin (TMPD) plus ascorbate (complex IV supported respiration). Importantly, these respiration rates can be partially restored by externally added cytochrome *c*, clearly indicating its release [18]. These data, together with confocal mitochondrial imaging, show that myocardial CIR leads to multiple types of mitochondrial damage and heterogeneous cytochrome *c* release. Importantly this damage correlated well with the decrease in heart contractile function. Oxidative stress during reperfusion/reoxygenation leads to peroxidation of proteins and phospholipids and in particular, to peroxidation of cardiolipin in the IMM. Phospholipid cardiolipin is reported to be essential for the normal function of various mitochondrial enzymes like cytochrome *c* oxidase, adenine dinucleotide translocase (ANT), mitochondrial creatin kinase (mitCK), and complex III. Importantly, the localized destruction of cardiolipin at the sites of free radical production would explain the different sensitivities of the different respiratory complexes to the same source of damage [125,126,127]. For example, diminished cardiolipin content in cardiac mitochondria due to peroxidation by ROS can lead to significant inhibition of cytochrome *c* oxidase [127]. Development of oxidative stress plays a significant role in the aging process [122] and in mechanisms of inborn (genetic) defects of mitochondrial complexes [124]. Alterations of mitochondrial function due to the damage to the mitochondrial respiratory complexes I, III and IV which decreases their activities. Also, alterations in mitochondrial membranes can cause an inhibition of several transport systems, uncoupling of mitochondrial respiration from OXPHOS and may further lead to the loss of certain mitochondrial components like enzymes of the intermembrane space and the matrix of mitochondria, together with cytochrome c release. Depletion of mitochondrial pulls of ATP/ADP and NADH/NAD are well documented [18,123,131]. However, less is known about consequences for the metabolic channeling, intracellular compartmentalization and cellular–mitochondrial integrations, including disruption of mitochondria–cytoskeleton interactions [119,135].

## 9. The Role of PTP Opening in Cardiac IR Injury


*Mitochondrial PTP Opening is a Critical Factor in IR Injury*


Multiple experimental studies provide evidence that mitochondrial PTP opening is convincingly involved in the pathogenesis of cardiac IR [136,137,138,139,140,141,142,143] and can be targeted to attenuate reperfusion-induced damage to the myocardium [138,139]. Patch-clamp studies on mitoplasts demonstrated that PTP are the non-specific channels localized in the IMM [140]. Opening of the pores increases colloidal osmotic pressure in the matrix and thereby induces swelling of mitochondria which, in turn, leads to ΔΨ_m_ loss, uncoupling of OXPHOS from respiration, and ROS overproduction. The massive Ca^2+^ release from mitochondria can result in cardiomyocyte hyper-contracture and cell death [28] in the heart. PTP opening has been shown to promote cell death through necrosis [141], although whether the cell dies through apoptosis or necrosis depends on the ATP availability (the number of mitochondria that undergo PTP opening). Thus, reperfusion injury causes mitochondrial dysfunction through increases in intracellular and mitochondrial Ca^2+^ and ROS, IMM depolarization and PTP opening [136,137,138,142,143,144] where Ca^2+^ increase and ROS overproduction cooperate to activate PTP opening. These effects, however, may be different in cardiac SSM and IFM. It has been demonstrated that cyclosporin A (immunosuppressive drug and a PTP inhibitor) at low concentrations may reduce IR injury in isolated cardiomyocytes and in Langendorff-perfused rat hearts [144,145,146,147]. Also, selective elimination of damaged mitochondria via autophagy (mitophagy) may be involved to maintain mitochondrial quality control in the cell. Moreover, mitoROS and the release of mitochondrial and cell content may result in activation of the inflammatory response with further damaging effect [133,147]. Studies on PTP opening and the effect of cyclosporin A in subcellular mitochondrial populations showed that IFM are more resistant to high Ca^2+^ [148].

The molecular identity of the PTP complex remains unidentified. Pioneering studies in this area identified VDAC and adenine nucleotide translocase (ANT) as the main proteins involved in the PTP complex. However, since 2004, several studies using genetic manipulations in mice and cells revealed that PTP opening occurs in the absence of these proteins, suggesting that they are not involved in the PTP complex and apparently play a regulatory role in pore formation (reviewed in [149,150,151,152,153,154]). The main positive modulator of the PTP is cyclophilin D, a cis-trans isomerase with a chaperone localized in the matrix. Pharmacological inhibition of cyclophilin D by cyclosporin A and sanglifehrin A has been shown to exert cardioprotective effects against IR injury in animal models of heart IR and in patients [155,156,157,158,159].

## 10. Possible Cardioprotective Strategies and Pharmacological Interventions

 *Several Cardioprotective Approaches and Specific Substances Can be Used to Reduce IR Injury*

Currently, therapeutic strategies for the treatment of cardiac IR through targeting mitochondria are mainly focused on the prevention of mitochondrial ROS production and Ca^2+^ overload [160]. Therefore, inhibition of excessive mitochondrial swelling through Ca^2+^-induced PTP opening (cyclosporin A and other inhibitors) can be considered to be one of the promising therapeutic strategies in the reduction of cardiac IR injury. It has been suggested that suppression of mitochondrial respiratory chain activity (e.g., complex I inhibition) during ischemia can to some degree decrease ROS and may thus be protective, but this inhibition must be reversible [123]. Various antioxidants like alpha-tocopherol, coenzyme Q10 and α-lipoic acid [161] and, in particular, mitochondria-targeting drugs such as melatonin [162], polyphenols [163], idebenone derivative of targeted to mitochondria triphenylphosphonium cation (ubiquinonyl) decyltriphenyl-phosphonium bromide, MitoQ) and Skulachev developed lipophilic cation (SkQ) [161] and mitochondria-targeting glutathione (mitoGSH) [164] are shown to have a capable cardioprotective effects in several models. Also, manganese superoxide dismutase (MnSOD) overexpression or mimetics, some cell/mitochondria-permeable antioxidant peptides [165,166] and Ca^2+^ antagonists or chelators like cell-permeable analogs of ethylene-diamine-tetraacetate (EGTA) have been elucidated extensively to prevent IR injury and its consequences [35,123,167]. 

Another approach for cardioprotection by the prevention of oxidative stress can be mild uncoupling of mitochondria [168,169] chemically (e.g., by dinitrophenol [170] and propofol [171]), or via overexpression of specific mitochondrial uncoupling proteins (e.g., UCP2 or UCP3), both leading to a decrease of ΔΨm and, therefore, reducing the intensity of superoxide production by mitochondrial respiratory chain complexes [172,173]. In addition, a new methodology of mitochondrial transplantation (by injection) in the IR injury of the myocardium, to support heart ATP level and thus cardiac contractile function, has been proposed [174]. However, many questions remain and the benefit of mitochondrial transplantation in clinics is still obscure [175].

### 10.1. Protection Against Cardiac IR Injury by Ischemic Preconditioning


*Preconditioning can be Considered as an Important Strategy in IR Injury*


Ischemic preconditioning (IPC) induced by several brief (3–5 min) episodes of ischemia and reperfusion prior to sustained ischemia has been recognized as a promising therapeutic strategy for the treatment of cardiac IR injury [176,177,178,179,180,181,182,183]. It has been proposed that ROS produced by mitochondria and several specific signaling pathways play a significant role in the cardioprotective effects of IPC [184]. For example, mild oxidative stress by heart perfusion with H_2_O_2_ in lower concentrations may have some protective effects, thus simulating the effects of IPC. The cardioprotective effects of IPC have been shown to be mediated through the inhibition of PTP opening during cardiac IR. Also, it has been demonstrated that protein kinase C [184] with the involvement of mitochondrial ATP-sensitive potassium channels [185,186] and several other signaling kinases can play an important role in the mechanisms involved in IPC and protection against heart IR. However, the detailed mechanisms and links between IPC and protection against heart IR needs further analysis. 

Therefore, various cardioprotective approaches and drugs that protect mitochondrial function, structure stability, complex dynamics and turnover, including various mitochondrial antioxidants and uncouplers and ischemic pre- and post-conditioning, can be considered as the main strategies to protect mitochondrial and cardiovascular function, and thus enhance longevity.

## 11. Intracellular Energy Transfer and its Changes in Cardiac IR: Creatine-Phosphocreatine Shuttle


*Alterations in Mitochondrial Creatine Kinase and Intracellular Energy Transfer are Found in IR Injury*


Mitochondrial injuries are implicated in intracellular signaling and mitochondrial respiratory function plays a central role in cellular energy metabolism and redox regulation, particularly in the heart as a continuously active tissue which depends on aerobic energy supply. Studies of the delicate bioenergetic mechanisms in the heart have demonstrated a key role for the mitochondrial creatine kinase (mitCK) for metabolic channeling and intracellular micro-compartmentalization, and resulted in the discovery of mitochondrial functional complexes with other cellular organelles, such as myofibrils and the sarcoplasmic reticulum, forming intracellular energetic units [83,187,188]. Importantly, mitCK and, in particular, its functional links with energy transferring systems [189,190,191], can be very sensitive to cardiac ischemia (due to increases in cellular inorganic phosphate level) and various cardiomyopathies [12,192,193,194,195]. Moreover, the mitCK system can be damaged by oxidative stress, due to possible oxidation of the enzyme-essential –SH residues by ROS [196]. A detailed analysis of mitochondrial respiratory function (ADP kinetics) and coupled mitCK systems in permeabilized fibers from different muscles (heart, quadriceps, gastrocnemius) of creatine kinase knockout mice revealed mitochondrial remodeling with subsequent effects on metabolic channeling, most probably as an adaptive response to the lack of creatine kinase [197,198]. Moreover, significant changes in the coupled mitCK system and mitochondrial remodeling have been demonstrated in various pathologies including IR injury [193], heart failure [12,192] and various cardiomyopathies [194,195]. Also, mitCK can protect mitochondria against PTP opening and depolarization [199]. Alterations in mitochondrial energetics, micro-compartmentation of adenine nucleotides and cellular energy transfer play a pivotal role in the mechanisms and pathophysiology of heart IR. 

Therefore, the mitCK system and the creatine–phosphocreatine energy transferring shuttle may be considered as additional important targets for protective mediation in cardiac IR injury [200].

## 12. Conclusions

In summary, mitochondrial damage and dysfunction are essential in the molecular mechanisms leading to IR injury of the heart. The scientific information obtained from mitochondrial physiology research can be useful for basic and clinically oriented studies, as well as for the development of new diagnostic approaches and tests for cardioprotection strategies. Also, cardioprotective interventions that can modulate mitochondrial dynamics/turnover and autophagy may be useful to improve energy metabolism and cardiovascular function after IR and enhance longevity. Moreover, a better understanding of the molecular mechanisms responsible for mitochondrial damage in CIR may provide the basis for interventional strategies aimed at the improvement of heart preservation in organ transplantation, thus enhancing organ recovery. A detailed characterization of the molecular mechanisms implicated in mitochondrial physiology and pathology will certainly help in the development of several new therapeutic approaches.

## Figures and Tables

**Figure 1 antioxidants-08-00454-f001:**
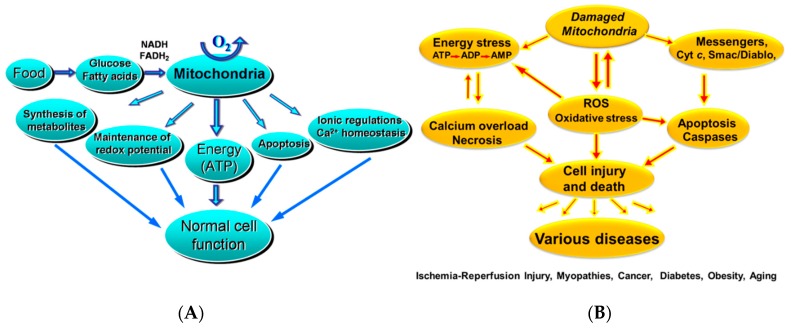
(**A**) The roles of mitochondria in normal cell function; and (**B**) in various cell damage/injuries. Mitochondrial function and dysfunction contribute to cell viability and injury by several mechanisms. ROS—reactive oxygen species.

**Figure 2 antioxidants-08-00454-f002:**
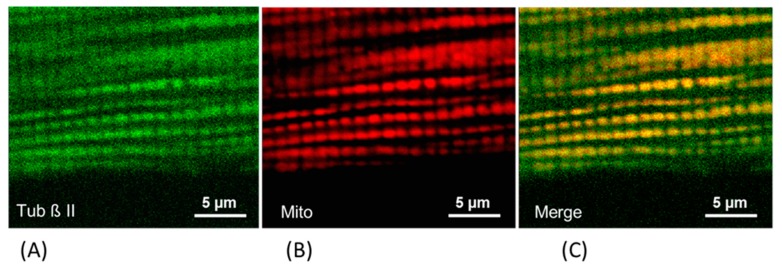
Fluorescence confocal evidence for the co-localization of the (**A**) tubulin beta II (Tub β II) (visualized by the specific antibodies against tubulin beta II, green), with mitochondria (Mito) (**B)** visualized by the specific mitochondrial potential-sensitive probe tetramethylrhodamine methyl ester (TMRM, red). Merged image (**C**).

**Figure 3 antioxidants-08-00454-f003:**
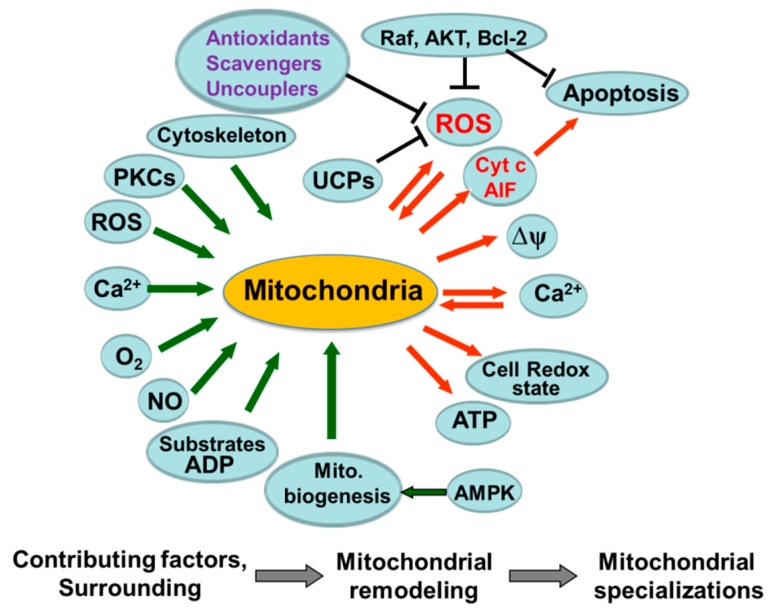
A scheme summarizing hypotheses regarding the possible origin and mechanisms contributing to the heterogeneity of mitochondria and mitochondrial function. Complex communications of mitochondria with a cell at rest and factors which can be involved in the formation of mitochondrial heterogeneity are shown. ΔΨ—mitochondrial potential. AMPK—AMP-activated protein kinase; PKC—protein kinase C; UCPs—uncoupling proteins; AIF—apoptosis- inducing factor; Raf—rapidly accelerated fibrosarcoma (RAF protein kinases); AKT—Protein kinase B (Akt, serine/threonine protein kinase); Bcl-2—(B-cell lymphoma 2 protein) antagonist of cell death.

**Figure 4 antioxidants-08-00454-f004:**
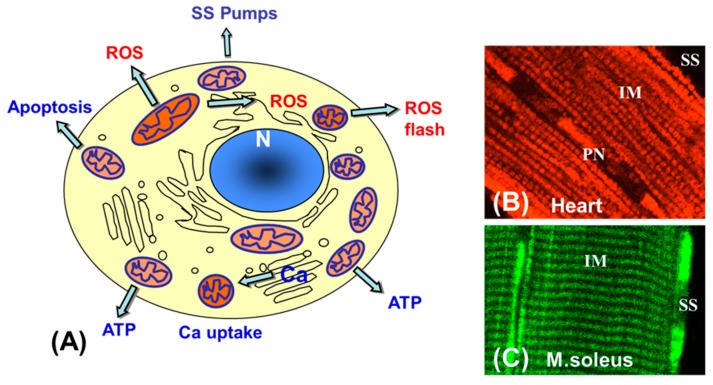
Mitochondrial heterogeneity and subpopulations. Mitochondrial subsets may have different region-specific specializations depending on their intracellular localization and environment (**A**). Mitochondrial subpopulations in a cardiac cell: SS—subsarcolemmal, IM—intermyofibrillar and PN—perinuclear mitochondria visualized by TMRM (red) (**B**). SS (subsarcolemmal) mitochondrial clusters in soleus muscles (M. soleus) visualized from the auto-fluorescence of mitochondrial flavoproteins, fluorescent in their oxidized state (green) (**C**).

**Figure 5 antioxidants-08-00454-f005:**
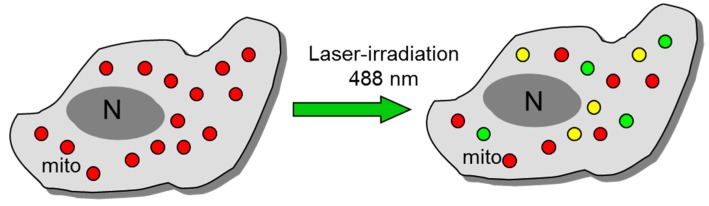
Laser irradiation as a tool for mitochondrial production of ROS. Note a significant heterogeneity of mitochondria in the cell in relation to mitoROS levels and degrees of mitochondrial depolarization (decline in the inner-membrane potential). mitoROS were visualized with 2,7-dihydrodichlorofluorescein (DCF) by 488 nm laser irradiation. Mitochondrial membrane potential was monitored with tetramethylrhodamine methyl ester (TMRM) by simultaneous 543 nm laser irradiation.

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
