# Peer review of "The Role of Mitochondria in the Mechanisms of Cardiac Ischemia-Reperfusion Injury"

_antioxidants, 2019, doi:10.3390/antiox8100454_

Round 1
Reviewer 1 Report
The manuscript “Mechanisms of Mitochondria-Mediated Cell Death in Cardiac Ischemia-Reperfusion Injury” by Kuznetsov A et al. has been submitted by the authors as a review to discuss in extensive manner the role of mitochondria in cardiac IR injury and in the cell death related to the cardiac damage, as suggested by the title of the paper.
-The authors describe extensively the mitochondrial structure, function, dynamics and heterogeneity in physiological and some pathological cellular conditions, but only in the last sections of the manuscript they discuss the mechanisms activated by mitochondrial dysfunction in the pathogenesis of cardiac IR injury and in the activation of cell death program. The authors should summarize the first sections of the manuscript and amply the section concerning the molecular mechanisms involved in cardiac IR damage to the myocardium. Furthermore, they should also amply the paragraph that describe the new therapeutic strategies for treatment of cardiac IR-mediated damages through targeting mitochondria.
The authors should also improve the manuscript in the different paragraphs with a more extensive description of innovative studies published on the proposed topic.
Author Response
Rev#1
The manuscript “Mechanisms of Mitochondria-Mediated Cell Death in Cardiac Ischemia-Reperfusion Injury” by Kuznetsov A et al. has been submitted by the authors as a review to discuss in extensive manner the role of mitochondria in cardiac IR injury and in the cell death related to the cardiac damage, as suggested by the title of the paper.
The authors describe extensively the mitochondrial structure, function, dynamics and heterogeneity in physiological and some pathological cellular conditions, but only in the last sections of the manuscript they discuss the mechanisms activated by mitochondrial dysfunction in the pathogenesis of cardiac IR injury and in the activation of cell death program. The authors should summarize the first sections of the manuscript and amply the section concerning the molecular mechanisms involved in cardiac IR damage to the myocardium. Furthermore, they should also amply the paragraph that describe the new therapeutic strategies for treatment of cardiac IR-mediated damages through targeting mitochondria. The authors should also improve the manuscript in the different paragraphs with a more extensive description of innovative studies published on the proposed topic.
We thank to the Reviewer for his Comments which helped us in improving of the revised version of the MS.
We believed that more overall knowledge about mitochondria can be useful for the understanding their roles and changes in pathology. The problem of mitochondrial heterogeneity thought to be important, since different mitochondrial subsets may have different sensitivities and roles in pathology.
However, we are fully agreed with Reviewer. The MS was substantially revised (marked in red) and restructured in the way of Reviewer Recommendations.
The part and sections of the MS regarding general principles of mitochondrial physiology were largely compressed and shortened. Instead, more information was added about mitochondrial dysfunction in the pathogenesis of IR injury and, in particular, about cardioprotective strategies, which now organized as a separate section.
Accordingly, a number of new, relevant and recent references were added.
So, we consider now our revised MS as more logically organized and basically improved. Thank you again for your evaluation and assistance.
Reviewer 2 Report
A better understanding of the molecular mechanisms responsible for mitochondrial damage in Cardiac ischemia-reperfusion is very important, and this article is a large and comprehensive review and may offer a new perspective. However in my opinion several points need to be improved.
General comments
Abstract
The objective it has to be clearer. The authors could talk a little more about each point that will be addressed.
Introduction
1 - The general structure is unclear. In some cases there is point 2, but then there is no point 3, only point 3.1 and 3.2. Please add missing points 3, 5, 6 and 8. Authors should make a brief introduction (may be only one line).
2 – Between the lines 185 and 202, there is no references. Since these lines are not summary and they are almost the totality of this point. Pease reformulate these sentences.
3 – Some figures are not visible, pleased improved the figures definition.
4 - In the legends of figures, abbreviations need to be defined.
5 – line 626 Fig.3 replace for figure 3,andexplain the phrase "these organelles .... cascades.
5 – The conclusion is very good.
Author Response
Rev#2
A better understanding of the molecular mechanisms responsible for mitochondrial damage in Cardiac ischemia-reperfusion is very important, and this article is a large and comprehensive review and may offer a new perspective. However in my opinion several points need to be improved.
General comments
Abstract
The objective it has to be clearer. The authors could talk a little more about each point that will be addressed.
We thank to the Reviewer for his Comments and assistance which were helpful for the MS improving.
Some revision of the Abstract has been done for its better clarity. Unfortunately, it is impossible to add here significantly more text (about each point) due to strong size limitations (200 words only).
Introduction
1 - The general structure is unclear. In some cases there is point 2, but then there is no point 3, only point 3.1 and 3.2. Please add missing points 3, 5, 6 and 8.
Thank you. It has been corrected.
Authors should make a brief introduction (may be only one line).
An introductory sentence (one line) was added to the each section, as well as we tried to make a short summary at the end.
2 – Between the lines 185 and 202, there is no references. Since these lines are not summary and they are almost the totality of this point. Pease reformulate these sentences.
The MS was largely restructured and reorganized. The sections concerning general mitochondrial physiology were compressed and shortened. The information about possible “cardioprotection” strategies and drugs is now organized as a separate section. Accordingly, more new references were added.
3 – Some figures are not visible, pleased improved the figures definition.
More explanatory text and information were added to the figure legends.
4 - In the legends of figures, abbreviations need to be defined.
Thank you. All abbreviations in the legends have been defined.
5 – line 626 Fig.3 replace for figure 3, and explain the phrase "these organelles ....
The word “cascades” was removed.
This section describes only some (few) aspects of cellular signaling.
5 – The conclusion is very good.
Round 2
Reviewer 1 Report
The revised Manuscript is more improved than the previous version and well organized.
Author Response
The revised Manuscript is more improved than the previous version and well organized.
Thank you.
Reviewer 2 Report
This article was revised appropriately. However, some figures need minor improvements.
In the figure 1B, removed the “etc” and include the other pathologies.
The author need to also to include the abbreviation used in the figures, like for example IR-injury, ROS….
Author Response
In the figure 1B, removed the “etc” and include the other pathologies.
Thank you. The the “etc” was removed (etc was also removed everywhere in the text), and we included aging as other pathology, instead etc.
The author need to also to include the abbreviation used in the figures, like for example IR-injury, ROS….
Thank you. The abbreviation used in the figures (IR-injury, ROS, and others like TMRM - tetramethylrhodamine methyl ester; DCF - 2,7-dihydrodichlorofluorescein) are now included.